# Purinergic Receptor Activation Protects Glomerular Microvasculature from Increased Mechanical Stress in Angiotensin II-Induced Hypertension: A Modeling Study [note 1]

**DOI:** 10.3390/ijms26051928

**Published:** 2025-02-24

**Authors:** Owen Richfield, Ricardo Cortez, Supaporn Kulthinee, Martha Franco, L. Gabriel Navar

**Affiliations:** 1Bioinnovation IGERT PhD Program, Tulane University, New Orleans, LA 70118, USA; 2Department of Physiology and the Tulane Hypertension and Renal Center of Excellence, Tulane University, New Orleans, LA 70112, USA; 3Department of Mathematics, Tulane University, New Orleans, LA 70118, USA; rcortez@tulane.edu; 4Department of Medicine, Division of Nephrology and Hypertension, Vanderbilt University Medical Center, Nashville, TN 37232, USA; kulthinee.supaporn@vumc.org; 5Renal Pathophysiology Laboratory, Department of Nephrology, Instituto Nacional de CardiologÍa “Ignacio Chávez”, México City 14080, Mexico; francoguevara@gmail.com

**Keywords:** glomerulus, mathematical modeling, purinergic activation, mechanical stress

## Abstract

Angiotensin II (Ang II)-induced hypertension increases afferent (AA) and efferent (EA) arteriole resistances via the actions of Ang II on the AT1 receptor. In addition to the increased interstitial levels of Ang II, the increased arterial pressure increases interstitial ATP concentrations. In turn, ATP acts on the purinergic receptors P2X1 and P2X7 to constrict the AA, preventing increases in plasma flow and single-nephron GFR (SNGFR). While the hemodynamic effects of P2 activation have been characterized, the resulting increases in mechanical stresses (shear stress and circumferential hoop stress) on the glomerular microvasculature have not been quantified. A mathematical microvascular hemodynamic glomerular model was developed to simulate blood flow and plasma filtration in an anatomically accurate rat glomerular capillary network. AA and EA resistances were adjusted to match glomerular hemodynamic data for control, Ang II-induced hypertension, and P2X1-blocked conditions. A blockade of the purinergic receptors reduced both afferent and efferent resistances, maintaining glomerular pressure at hypertensive levels but increasing blood flow and sheer stress significantly. Because glomerular pressure was maintained, hoop stress barely changed. Our results indicate that the activation of the purinergic system protects the glomerular microvasculature from elevated shear stress caused by increased blood flow that would occur in the absence of purinergic stimulation.

## 1. Introduction

Hypertension is a leading contributor to kidney disease worldwide, and as rates of hypertension continue to rise, an increase in the incidence of chronic kidney disease has followed [1,2]. The inappropriate stimulation of the renin–angiotensin system is a common factor involved in the progression of hypertension, resulting in higher production of angiotensin II (AngII), with consequent increases in afferent (AA) and efferent arteriole (EA) resistances and glomerular pressure [3]. The role of AngII in the alterations of glomerular hemodynamics in hypertension has been described in detail and has fostered the development of numerous anti-hypertensive therapies targeting AngII formation and receptor interactions that reduce glomerular injury in hypertension. However, other pathways mediating the constriction of the AA and EA have only recently been characterized [4]. In particular, the role of purinergic receptors, including P1 and P2 purinoceptors, on AA and EA smooth muscle cells in modulating glomerular pressure and flow has been elucidated experimentally in the past two decades [5,6,7,8,9,10]. Models of AngII-induced hypertension show a marked increase in renal parenchymal ATP, leading to enhanced AA and EA resistance through interactions with smooth muscle cell purinergic P2X receptors [6].

P2X receptors, consisting of seven subtypes (P2X1-P2X7), are ligand-gated ion channels that activate within milliseconds of ATP binding [11]. Under normotensive conditions, P2X1 receptors (P2X1R) are expressed on the renal microvasculature [12]. Interstitial ATP activates P2X1R and elicits afferent arteriolar vasoconstriction in response to increases in renal perfusion pressure [6], suggesting that P2X1R contributes to mechanical stress responses. In an Ang II-dependent hypertension model, afferent vasoconstriction was mitigated by a P2X1R inhibitor, indicating that P2X1R contributes to the increased vascular resistance in Ang II-dependent hypertension [10]. This evidence supports the hypothesis that P2X1R exerts a dominant influence abrogating the actions of Ang II in chronic angiotensin II-dependent hypertension. This adaptation prevents high renal perfusion pressure-induced glomerular capillary injury. In addition, P2X1R is upregulated in Ang II-dependent hypertension [9], contributing further to the dominance of P2X receptor control of glomerular dynamics in hypertension.

While numerous biological and physiological mechanisms interact to promote glomerular injury in hypertension, recent studies on mechanical stresses (shear stress, circumferential ‘hoop’ stress, and resulting strain) on the glomerular cells have identified injury pathways for scientific study and therapeutic development [13,14,15,16,17,18]. Namely, it is hypothesized that mechanical stresses are enhanced in hypertension, leading to enhanced injury. To test this hypothesis, it is essential to quantify these mechanical stresses for the design of more accurate experiments, particularly those that utilize novel in vitro systems to better recapitulate the mechanical microenvironment of glomerular cells [19,20]. In a series of studies, we developed and utilized an anatomically accurate mathematical model of glomerular blood flow and filtration to estimate how mechanical stresses are spatially distributed throughout the glomerular capillary network under varied pathophysiological hemodynamic conditions [21,22,23]. Importantly, the results from these studies correlated with known forms of glomerulosclerosis that correspond to the disease states that we simulated [24,25].

In the present study, we use our mathematical model of glomerular filtration and mechanics to quantitatively determine the impact of acute P2X1 receptor-induced vasoconstriction on the mechanical stresses exerted on the glomerular cells during hypertension. Hemodynamic data from previous studies utilizing an AngII-induced rat model of hypertension [9] are used to (1) estimate the effect of P2X1 receptor blockade on AA resistance and glomerular pressure and (2) quantify the resulting mechanical effects. Importantly, our results indicate that P2X1 receptor activation in hypertension protects the glomerular cells from mechanical injury, such that P2X1 blockade may enhance the risk of mechanical injury.

## 2. Results

Using the parameters above, we evaluated the impact of the P2X1 blockade on the magnitudes of mechanical stress exerted on the glomerular endothelial cells (shear stress) and podocyte foot processes (hoop stress). In Figure 1, we show that AngII treatment without P2X1 blockade reduces shear stress and elevates hoop stress. Interestingly, the blockade of P2X1 via NF449 greatly enhances shear stress to similar levels, independent of AngII treatment. However, AngII and NF449 treatment show similar changes in hoop stress over control, with NF449 negligibly affecting hoop stress magnitudes in the AngII-induced hypertension model.

In Figure 2, we show the localized shear stresses exerted on each glomerular capillary in the network based on model simulations. As we have shown in previous studies, the advantage of modeling the glomerulus in an anatomically accurate manner is the ability to map mechanical stresses and other quantities, such as CSGFR, on the entirety of the glomerular network topology (Figure 2A). Importantly, the model indicates that shear stresses are higher in one lobule as compared to the other two and are elevated in the capillaries nearest to the efferent arteriole. By plotting individual capillary shear stresses with NF449 treatment against baseline shear stress, Figure 2B shows that NF449 treatment results in a significant increase in shear stress from baseline in all glomerular capillaries in both Sham and AngII conditions. Interestingly, the inhibition of P2X1 produces a larger effect under AngII conditions than in control conditions. On a per-capillary basis, capillaries that have a high baseline shear stress also have the largest change in shear stress when P2X1 is inhibited. Taken together, these results suggest that localized areas of the glomerular capillary network (i.e., near the efferent arteriole) are at increased risk of mechanical injury with P2X1 inhibition.

## 3. Discussion

In this modeling study, we used an anatomically accurate mathematical model of blood flow and filtration in a rat glomerulus to estimate the magnitudes of mechanical stress exerted on glomerular cells with and without P2X1-induced vasodilation in an AngII-induced model of hypertension. Compared to Sham, AngII treatment was associated with a reduction in shear stress and an elevated hoop stress on the glomerular capillaries. Hoop stress, responsible for the stretching of podocyte foot processes, is known to be deleterious to podocyte structural integrity: Biaxial strain injures podocytes in vitro, causing foot process effacement and reorganization of the actin cytoskeleton [26]. Shear stress causes the endothelial cell release of TGF-β1 in a magnitude-dependent manner [27], and TGF-β1 acts synergistically with stretch to damage podocytes [26] and increases mesangial cell proliferation and glomerulosclerosis [28]. Thus, changes in either the shear stress on the glomerular endothelium or the hoop stress on podocytes may contribute to the progression of glomerulosclerosis in hypertension.

Our results indicate that acute purinergic blockade, using NF449 to inhibit the P2X1 receptor on AA and EA SMCs, results in a substantial increase in shear stress on the glomerular endothelium but only alters hoop stress in Sham conditions, not in hypertensive conditions. Importantly, by modeling the glomerulus as a network of capillaries, we identified the capillaries that show the largest change in shear stress with the application of NF449: namely the capillaries that coalesce to form the EA. We have previously identified the capillaries most at risk of injury due to hoop stress, which are the capillaries branching off the AA. Importantly, the capillaries closest to the EA and AA are in anatomical association with one another at the vascular pole of the glomerulus. Our model results motivate the hypothesis that the increased shear stress on the efferent capillary endothelial cells causes TGF-β1 release, which further exacerbates the injury dealt to podocytes by elevated hoop stress on the afferent capillary walls. The modeling results indicate that by reducing shear stress on the glomerular endothelium, purinergic P2X1 receptor activation protects the glomerulus from mechanical stress in both normotensive and hypertensive conditions.

Importantly, the results of this study and the hypotheses they suggest must be qualified with the limitations in the scope of the study; while we particularly focused on acute P2X1 receptor blockade, this does not necessarily predict how chronic P2X1 receptor blockade would impact glomerular mechanics in Angiotensin II-induced hypertension. In the case of our modeling study, acute administration of the antagonist allowed for the evaluation of a direct effect of the antagonist on the renal microcirculation. This was suitable for our model use. With chronic administration of the P2X1 receptor antagonist, hemodynamic alterations may not be due only to changes in afferent and efferent arteriole resistances since the influence of decreased inflammatory and fibrotic responses induced by the P2X1 antagonist would differentially and independently impact glomerular hemodynamics. In this regard, T cells, neutrophils, and macrophages have P2X1 receptors on their surface, and the chronic administration of specific blockers would decrease the inflammatory reaction and the release of cytokines and TNFα [29,30,31].

A second limitation of our study is that we focused on P2X1 receptor blockade while multiple other subtypes of P2 receptors may also play a role in maintaining glomerular mechanics. The participation of P2X1, P2X4, and P2X7 receptors has been previously addressed with specific antagonists in Ang II-dependent hypertension. Glomerular hemodynamics in Ang II-induced hypertension show that P2X1R and P2X7R blockade reduce afferent and efferent arteriolar resistances, leading to increases in glomerular plasma flow, ultrafiltration coefficient, and single-nephron glomerular filtration rate to near-normal values. P2X7 blockade acts in a quantitatively similar manner to P2X1 blockade. P2X4 blockade does not modify baseline glomerular hemodynamics, but ivermectin, a P2X4-positive allosteric agonist, induces renal vasodilation [9]. Collectively, with our modeling results, we predict that P2X7 receptors similarly protect the glomerular vasculature from deleterious mechanical stress, while P2X4 receptor activation increases mechanical stresses in the glomerulus.

The use of the AngII-induced model of hypertension highlights the interaction between AngII and ATP in modulating AA and EA resistance. This interaction has been the focus of recent physiological studies using the juxtamedullary nephron preparation [10]. The complex interactions between Ang II and ATP occur under conditions of Ang II-induced hypertension. During the development phase, the actions of Ang II dominate to cause systemic and renal vasoconstriction, exerting constriction of the afferent and efferent arterioles and reductions in the glomerular filtration coefficient. In addition, the increase in arterial pressure leads to progressive increases in interstitial levels of ATP, which contribute further to the augmented renal vascular resistance. Although the AT1 receptors remain active, there is the post-receptor convergence of signaling pathways such that the powerful actions on the afferent arterioles elicited by the P2X1 receptor-mediated activation of calcium channels eventually dominate the vascular resistance during the sustained phase of hypertension.

We have previously developed a model of renal autoregulation to examine how the myogenic and tubuloglomerular feedback mechanisms interact to control glomerular mechanics with changes in perfusion pressure [22], but we have not incorporated AngII or ATP receptors nor antagonist pharmacology. Ideally, our model would build on previous work that modeled calcium dynamics in AA SMC to control AA resistance [32,33,34,35,36,37]. Since the interaction between ATP and AngII in AA SMC is calcium-dependent [10], it is possible that a model of AA SMC calcium dynamics, with ample modification, could predict the mechanism by which purinergic and AngII receptors interact. Incorporating new experimental data, our model will be used to delineate the role of each purinergic receptor subtype and AngII receptors in the progression of mechanical injury in the glomerulus in hypertension. This will support further understanding of these disease mechanisms and therapeutic avenues.

P2X7 antagonists are under development by several drug companies and have been tested in patients with inflammatory diseases such as rheumatoid arthritis, neuroinflammation, pain, and cancer [38]. The newly developed P2X7 antagonist AZD 9056, targeting Crohn’s disease, is such an example [39]. However, our modeling results indicate that, even under normotensive conditions, P2X1 and P2X7 receptors play a pivotal role in maintaining glomerular mechanical stresses at baseline. Thus, a P2X7 antagonist may have a deleterious effect on the glomerulus, whether or not the antagonist’s target is the kidney. Further development of our mathematical model would support (1) the evaluation of the efficacy of a P2 receptor antagonist in improving glomerular function in hypertension and (2) the estimation of the antagonist’s impact on glomerular mechanics.

## 4. Materials and Methods

The mathematical model used in this study extends the work of previous studies that modeled blood flow and filtration in an anatomically accurate glomerular capillary network [40]. Our model is novel in that it does not assume a linear pressure profile on the length of the glomerular capillary but instead calculates both the internal pressure of the capillary and the volume lost to filtration simultaneously. We briefly describe the governing model equations and algorithm and direct readers to our previous work [21] for a more detailed description. All model development was performed in R version 3.5.1, Vienna, Austria. 

### 4.1. Model Formulation

The anatomy of our model of the glomerulus is taken from previous studies detailing the fixation, sectioning, imaging, and reconstruction of a glomerular tuft isolated from a rat kidney [41], wherein each capillary segment is separately measured (length and diameter), and the overall network topology (namely the connections between capillaries) is quantified. To use these data in our model, the model glomerulus is represented as a graph with 320 segments (capillaries) and 193 nodes (points at which capillaries coalesce and/or bifurcate). The EA and AA are represented by resistors R_E_ and R_A_, respectively, and govern the distribution of pressure from upstream of the afferent resistor (equal to MAP) and downstream of the efferent resistor (set equal to 15mmHg). Blood plasma volume, erythrocyte volume, and plasma protein mass are conserved across the network, resulting in concentration of plasma proteins, and erythrocytes as plasma water is filtered. For nodes *i* and *j*, the pressure profile *p_ij_* on the length of the capillary connecting the two nodes is obtained by solving(1)$$\frac{d^{2}p_{ij}\left(x\right)}{dx}-{a_{ij}}^{2}p_{ij}\left(x\right)=-{a_{ij}}^{2}p_{BS},$$

For *p_BS_*, the Bowman’s Space pressure is set equal to 14 mmHg. The parameter *a_ij_* is calculated assuming Poiseuille flow conditions in estimating the capillary resistance, *R_ij_*:(2)$${a_{ij}}^{2}=\frac{R_{ij}}{R_{ij}^{f}{L_{ij}}^{2}}=\frac{128\mu_{ij}}{R_{ij}^{f}L_{ij}\pi {D_{ij}}^{4}},$$

Here, *µ_ij_*, *D_ij_*, and *L_ij_* are the blood viscosity, diameter, and length of capillary *ij*, respectively. The resistance to filtration, *R_ij_^f^*, is calculated iteratively to balance the internal pressure *p_ij_* with the loss of fluid to filtration. For the nth iteration,(3)$${\left(R_{ij}^{f}\right)}^{n}=\frac{\int_{0}^{L_{ij}}\left({\left(p_{ij}\left(x\right)\right)}^{n-1}-p_{BS}\right)dx}{k\pi L_{ij}D_{ij}\int_{0}^{L_{ij}}\left({\left(p_{ij}\left(x\right)\right)}^{n-1}-p_{BS}-{\left(\Pi_{ij}\left(x\right)\right)}^{n-1}\right)dx}$$
where the colloid osmotic pressure Π*_ij_* is calculated as a cubic function of the plasma protein concentration [42]. The hydraulic conductivity, *k*, determines the permeability of the capillary wall to water and is thus inversely proportional to the filtration resistance. The second quantity that is calculated iteratively is the blood viscosity, *µ_ij_*, which is calculated as a nonlinear function of the capillary diameter, the plasma protein concentration, and the hematocrit in the vessel [43]. The overall algorithm and iteration procedure is depicted in Figure 3.

The advantage of using an anatomically accurate model of glomerular filtration is the ability to calculate the localized (capillary-wise) filtration rate, denoted *CSGFR*, and mechanical stresses on the glomerular capillary walls. For capillary segment *ij*, *CSGFR* is calculated:(4)$$CSGFR_{ij}=\frac{\int_{0}^{L_{ij}}\left(p_{ij}\left(x\right)-p_{BS}\right)dx}{L_{ij}R_{ij}^{f}}.$$

The shear stress exerted on the glomerular endothelium, denoted *τ*, and the hoop (circumferential stretching force) exerted on the circumference of the capillary, denoted *σ*, are calculated using Poiseuille’s equation and Laplace’s equation, respectively:(5)$$\tau_{ij}=\frac{32\mu_{ij}\int_{0}^{L_{ij}}Q_{ij}\left(x\right)dx}{L_{ij}\pi {D_{ij}}^{3}}$$(6)$$\sigma_{ij}=\frac{D_{ij}\int_{0}^{L_{ij}}\left(p_{ij}\left(x\right)-p_{BS}\right)dx}{2L_{ij}t_{ij}}.$$

For *t_ij_*, the thickness of the glomerular capillary wall, and *Q_ij_*, the blood flow on the length of the glomerular capillary,(7)$$\frac{dQ_{ij}\left(x\right)}{dx}=-\frac{p_{ij}\left(x\right)-p_{BS}}{L_{ij}R_{ij}^{f}}.$$

Finally, to extend the above equations to a network of glomerular capillaries, we assume perfect mixing of plasma proteins, nonlinear distribution of erythrocytes at network nodes [44], and conservation of total blood volume at each node. For *J,* the set of nodes *j* connected to node *i*,(8)$$\underset{j\epsilon J}{\sum }Q_{ij}=0.$$

This facilitates the simultaneous estimation of pressures at all network nodes and can be performed iteratively until R^f^ and µ converge for every capillary in the network (Figure 3).

**Figure 3 ijms-26-01928-f003:**
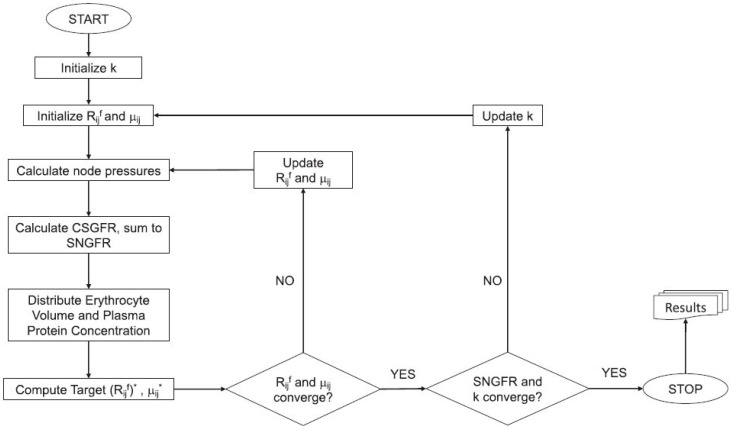
**Model algorithm.** Resistance to filtration, *R^f^*, and viscosity, µ, are iteratively calculated for each capillary segment until both quantities converge. After convergence is reached, SNGFR is calculated and compared to a target SNGFR value, and k is altered to better fit the model to the target SNGFR. Once all quantities have converged, the model is fit to the data, and individual glomerular capillary filtration rates (CSGFRs) and mechanical stresses can be assessed. In Table 1, we perform a dimensional analysis of the relevant model variables.

**Table 1 ijms-26-01928-t001:** A dimensional analysis of relevant variables to the model. Each variable’s reported units are converted to a set of working units represented by symbols: M is mass in µg, T is time in s, and L is length in µm. Working units for L and T are used based on data from the source literature, while the working unit for M (µg) allows for each variable to be sufficiently large to avoid computational error associated with operating on small digits (<10^−14^). Conversion to working units involves the multiplication of the reported units by the conversion factor listed.

Variable	Reported Units	Symbolic Units (M, T, L)	Conversion to Working Units (µg, s, µm)
Diameter, D	µm	L	1 µm
Length, L	µm	L	1 µm
Pressure, p	mmHg	M T^−2^ L^−1^	133,322 µg s^−2^ µm^−1^
Flow, Q	nL/min	L^3^ T^−1^	16,666.67 µm^3^ s^−1^
Resistance, R	mmHg/nL/min	M T^−3^ L^−4^	8 µg s^−3^ µm^−4^
Shear Stress, τ	dynes/cm^2^	M T^−2^ L^−1^	100 µg s^−2^ µm^−1^
Hoop Stress, σ	kPa	M T^−2^ L^−1^	1,000,000 µg s^−2^ µm^−1^
Viscosity, µ	cP	M T^−1^ L^−1^	1 µg s^−1^ µm^−1^
CSGFR	nL/min	L^3^ T^−1^	16,666.67 µm^3^ s^−1^
Hydraulic Conductivity, k	nL/min/mmHg/ µm^2^	M^−1^ T^3^ L^2^	0.125 µg^−1^ s^3^ µm^2^

### 4.2. Simulating AA and EA Purinergic Receptor Blockade

To evaluate the impact of EA and AA purinergic activation on the mechanical stresses in the glomerulus, we fit the glomerulus model described above to data from glomerular micropuncture studies wherein the P2X1 receptor blocker, NF449, was administered to rats with and without AngII-induced hypertension [9]. It should be pointed out that an acute infusion of the P2X1 receptor blocker was chosen to prevent further effects of the compound in the tubulointerstitial inflammation developed in this model since they could influence glomerular hemodynamics [29]. Sham animals were controls that underwent glomerular micropuncture but received no pharmacological treatment. Each case (Sham, Sham + NF449, AngII, and AngII + NF449) was recreated by assuming the MAP from the source literature as the input pressure and then altering the afferent and efferent resistances (R_A_ and R_E_, respectively) until the plasma flow Q_A_ and average glomerular capillary pressure P_GC_ matched hemodynamic data from the source literature. The hydraulic conductivity k was iteratively optimized to recreate the SNGFR from source literature, as described in Figure 3. Importantly, afferent and efferent resistances could be altered separately from k because we have previously shown that changes in k produce negligible alterations in P_GC_ and Q_A_ [21]. Results of our model fit are shown in Table 2.

## Figures and Tables

**Figure 1 ijms-26-01928-f001:**
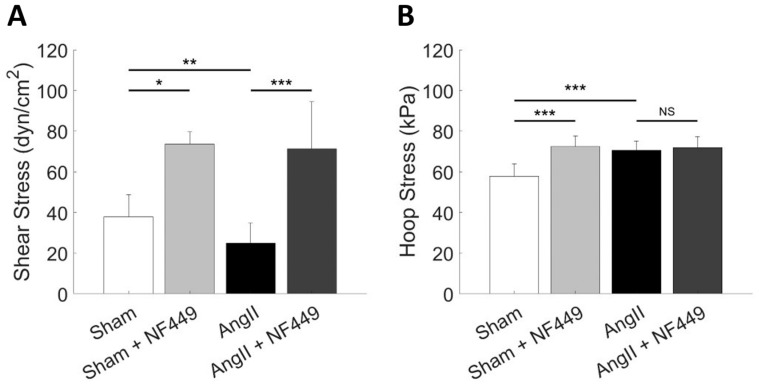
**Average mechanical stresses exerted on the glomerular capillary network.** (**A**) Shear stress on the endothelial cells is elevated with blockade of the P2X1 receptor, consistent with increased flow under these conditions. (**B**) Circumferential hoop stress is elevated in all treatment groups, independent of P2X1 blockade. * *p* < 0.05; ** *p* < 0.005; *** *p* < 0.0001; NS, not significant, as assessed by Welch’s *T*-test.

**Figure 2 ijms-26-01928-f002:**
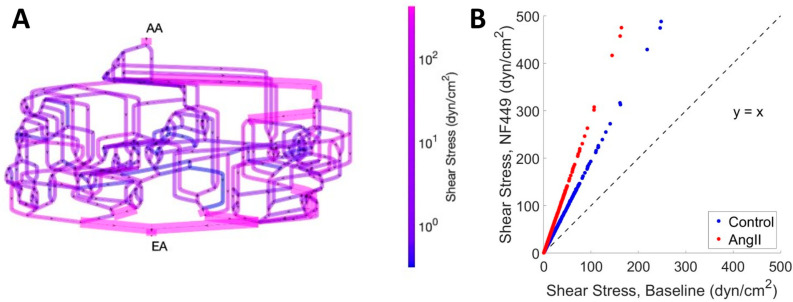
**Localized shear stresses exerted on each glomerular capillary.** (**A**) Shear stress values are mapped on their corresponding capillaries in the glomerular capillary network. Each segment of the network represents a capillary segment, with nodes representative of points of coalescence and/or bifurcation. Arrows indicate flow direction of the model. AA: afferent arteriole, EA: efferent arteriole. Diagram not drawn to scale. (**B**) Individual capillary segment shear stresses are plotted for the NF449 treatment groups against their respective control (AngII or Sham), demonstrating significant differences between capillaries of each case and the same capillaries between cases.

**Table 2 ijms-26-01928-t002:** Simulations based on hemodynamic data generated from previous glomerular micropuncture studies [9]. P_GC_: glomerular capillary pressure, Q_A_: glomerular plasma flow, SNGFR: single-nephron glomerular filtration rate, MAP: mean arterial pressure, R_A_: afferent arteriole resistance, R_E_: efferent arteriole resistance, k: glomerular capillary hydraulic conductivity. MAP was taken from the source material [9]. R_A_, R_E_, and k were altered to fit the P_GC_, Q_A_, and SNGFR reported in the source material.

Condition	P_GC_ (mmHg)	Q_A_ (nL/min)	SNGFR (nL/min)	MAP (mmHg)	R_A_ (mmHg min/nL)	R_E_ (mmHg min/nL)	k (×10^−5^ nL/min/µm^2^/mmHg)
Sham	46.9	125.8	38.8	119.7	2.9	1.3	1.83
Sham + NF449	56.2	215.6	53.7	122.8	1.5	0.9	1.82
AngII	55.8	76.1	21.3	162.8	6.6	3.0	0.71
AngII + NF449	56.6	213.1	48.8	171.6	2.5	1.0	1.53

## Data Availability

The data and mathematical models used to support the findings of this study were coded in R version 3.5.1 and are available on GitHub at https://github.com/omrichfield/autoreg_glommod, last accessed on 19 February 2025, and on Zenodo at https://doi.org/10.5281/zenodo.11114851.

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
