# Peer review of "Purinergic Receptor Activation Protects Glomerular Microvasculature from Increased Mechanical Stress in Angiotensin II-Induced Hypertension: A Modeling Study"

_ijms, 2025, doi:10.3390/ijms26051928_

Round 1

Reviewer 1 Report

Comments and Suggestions for Authors

Summary
The authors provided a well detailed mathematical model of for simulating the blood flow and plasma filtration in a rat glomerular capillary network as consequence of the purinergic activation. Authors should address a few points listed in the following review report. The manuscript is well-constructed, adherence to MDPI’s manuscript template, including line numbering, is required to meet publication standards.

Abstract

  • To improve readability, consider splitting the following sentence:
    “In addition to the increased interstitial levels of Ang II, the increased arterial pressure increases interstitial ATP concentrations which act on the purinergic receptors P2X1 and P2X7, to constrict the AA, preventing increases in plasma flow and single nephron GFR (SNGFR).”

Introduction

  • While concise, the introduction would benefit from a deeper exploration of the role of purinergic neurotransmission in modulating extracellular ATP. Specifically, highlight how the protein structure of purinergic receptors—composed of distinct functional domains—plays a role in these mechanisms. Gene ontology terms could provide further insight. I recommend the following references for support: https://doi.org/10.1177/2398212818817494; https://doi.org/10.1016/j.biopha.2022.114205
  • Additionally, consider discussing genetic variants in purinergic receptor gene families and their implications for ion channelopathies affecting muscular and skeletal systems. These references may be helpful: https://doi.org/10.3390/cimb46020073; https://doi.org/10.1016/j.nmd.2020.12.003
  • Introduce the aim of your study in a separate paragraph for clarity:
    “In the present study, we utilized this mathematical model to [state the specific aim].”

Materials and Methods

  • While the mathematical model is well-explained, consider including the units of measurement and dimensional analysis for each variable analyzed. This addition will enhance the clarity and scientific rigor of the methodology.

Results

  • For Figure 2, consider including standard deviations and the results of statistical tests, such as Duncan’s test or ANOVA, to evaluate the significance of observed differences. If additional replicates are available, this analysis could strengthen your conclusions.

Discussion

  • Discuss any limitations encountered with the mathematical model and how these could be addressed in future work.
  • Briefly describe how the development of an accurate model could pave the way for novel therapeutic strategies.

Author Response

Thank you for your comments and suggestions. We have attached a point-by-point response to your comments.

Reviewer 2 Report

Comments and Suggestions for Authors

Dear authors,

1.        The authors evaluate the effects of acute P2X1 receptor blockade. However, the long-term effects and the role of P2X1 in chronic hypertension have not been discussed.

2.        While the focus is on the P2X1 receptor, the roles of other purinergic receptors, including P2X7, have not been sufficiently addressed. Discussing these receptors will provide a more comprehensive understanding of how they influence mechanical stress and damage in the glomeruli.

3.        The authors need to further describe the mechanisms underlying the interactions between AngII and ATP and their effects on the resistance of AA and EA. This will help readers gain a deeper understanding of how P2X1 receptor activation can prevent glomerular damage associated with hypertension.

4.        While the authors consider the impact of P2X1 receptor blockade on mechanical stress in the glomeruli, it would be helpful to discuss how the results may translate into actual treatments, as well as the potential synergistic effects when combined with other therapies. This would clarify the clinical applicability and make it easier for readers to understand.

Author Response

(The authors gave the same response as above.)

Round 2

Reviewer 1 Report

Comments and Suggestions for Authors

Authors addressed all  reviewer's comments